# Divergent access to 5,6,7-perifused cycles

Jingpeng Han[1], Yongjian Yang[1], Yingjian Gong[1], Xuan Tang[1], Yi Tian[1] & Baosheng Li [1]✉

Nitrogen-containing heterocycles are the key components in many pharmaceuticals and functional materials. In this study, we report a transition metal-catalyzed high-order reaction sequence for synthesizing a structurally unique N-center 5,6,7-perifused cycle (NCPC). The key characteristics include the formation of a seven-membered ring by the 8π electrocyclization of various alkenes and aromatic heterocycles as π-components, in which metal carbene species are generated that further induce the cleavage of the α-C-H or -C-C bond. Specifically, the latter can react with various nucleophilic reagents containing -O, -S, -N, and -C. The stereo-controlled late-stage modification of some complicated pharmaceuticals indicates the versatility of this protocol.

Natural products are often used as the lead compounds[1-5] in designing or searching for pharmaceuticals. For example, 5,6,7-perifused carboncycle compounds, tricyclic compounds that share the same center atom, form the core skeleton in many bioactive natural products (Fig. 1a)[6-9]. These scaffolds might be used as the lead structures for guiding the synthesis of analogs, thus, promoting drug development. In contrast, N-center perifused cycles (NCPC) might further reveal interesting perspectives for fundamental studies and technological applications because the nitrogen atom has higher electronegativity and smaller van der Waals radius than the carbon atom, which can increase molecular polarity and water solubility[10,11], thus improving molecular selectivity[12] and availability (Fig. 1b). However, when the nitrogen atom is located in the central position of the tricyclic system, the valence state and non-bonding ion pairs can increase the difficulty of synthesis. Specifically, when the medium ring is involved, the difficulty in synthesis might be further increased due to its high torsional strain, unfavorable entropic and enthalpic factors[13-18], and other reasons. Previous routes for synthesizing NCPC start from α-formyl-pyrrole[19-22] or -pyridine[23] as raw materials involved a cumbersome multi-step strategy, and the related methods were also rare (Fig. 1c). In some contexts, the lack of a simple catalytic protocol to access skeletons might also hinder their applications in synthetic chemistry. Therefore, an efficient synthetic method needs to be developed to further extract or elucidate their application in organic synthesis.

Transition metal-catalyzed cascade cyclization[24-26] represents an efficient strategy for assembling various polycyclic systems. As shown in Fig. 1d, we speculate that the metal-catalyzed 5-endo-cyclization of pyridine-alkyne might form zwitterionic vinyl metal species[27-40] that might resonate to become a reactive conjugated 8π electronic system and a metal carbene[41-43]. Electrocyclization reactions[44-49] can be driven by heating and are not affected by external conditions. However, the reactivity of metal carbene is closely related to the reaction system. Thus, electrocyclization might be conducted for constructing seven-membered ring structures, promoting the formation of NCPC metal carbene intermediates that provide a great opportunity for applying reactive metal carbene species as a common intermediate for the divergent synthesis of substituted NCPC with high efficiency and flexibility. The metal carbene is a highly versatile intermediate, and NCPC metal carbene provide synthetic chemists access to various structural motifs. The present method might be a suitable alternative for synthesizing NCPC derivatives and could be a way to produce the related N-center substituted series of natural products, which cannot be prepared using other means.

In this work, we report the results of metal-catalyzed cascade cyclization for the divergent synthesis of NCPC. The protocol includes the following points. (1) The final seven-membered ring is constructed by 8π electrocyclization, which is interesting due to its value in addressing the differences in the formation of a medium ring in complicated polycyclic systems. (2) The process includes multi-chemical events, which help in constructing NCPC structural units directly and rapidly through a high-order reaction sequence. (3) The "non-classical" tricyclic metal carbene species are used as a common intermediate while synthesizing the NCPC. (4) The structurally unique NCPC might be used as a platform model for discovering some synthetically useful compounds. (5) This study might provide useful information for designing perifused ring analogs with potent activity.

[1]School of Chemistry and Chemical Engineering, Chongqing University, 174 Shazheng Street, 400044 Chongqing, P. R. China. ✉e-mail: libs@cqu.edu.cn

a) Representative perifused 5,6,7-tricarbocycles natural products

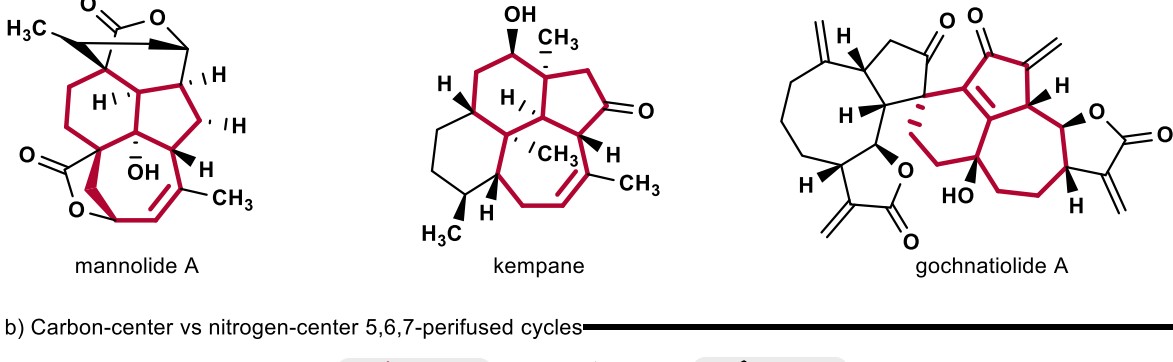

mannolide A                    kempane                    gochnatiolide A

b) Carbon-center vs nitrogen-center 5,6,7-perifused cycles

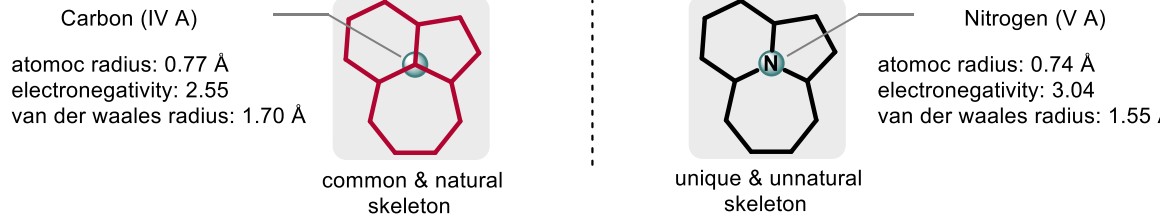

Carbon (IV A)

atomoc radius: 0.77 Å
electronegativity: 2.55
van der waales radius: 1.70 Å

common & natural
skeleton

Nitrogen (V A)

atomoc radius: 0.74 Å
electronegativity: 3.04
van der waales radius: 1.55 Å

unique & unnatural
skeleton

c) Synthetic routes for the N-center 5,6,7-perifused cycles (literatures)

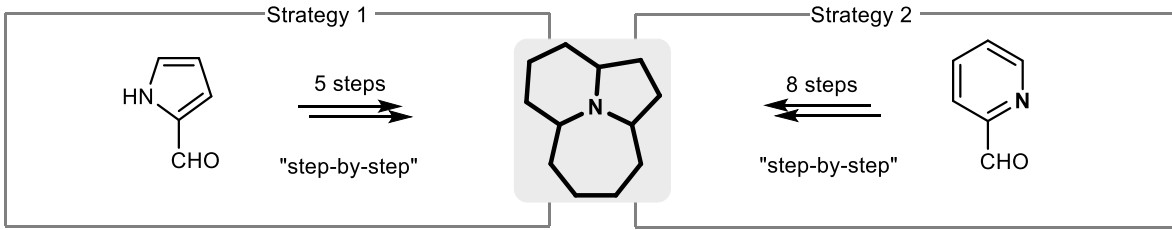

Strategy 1

5 steps

"step-by-step"

Strategy 2

8 steps

"step-by-step"

d) Metal catalyzed the divergent synthesis of N-center 5,6,7-perifused cycles (this work)

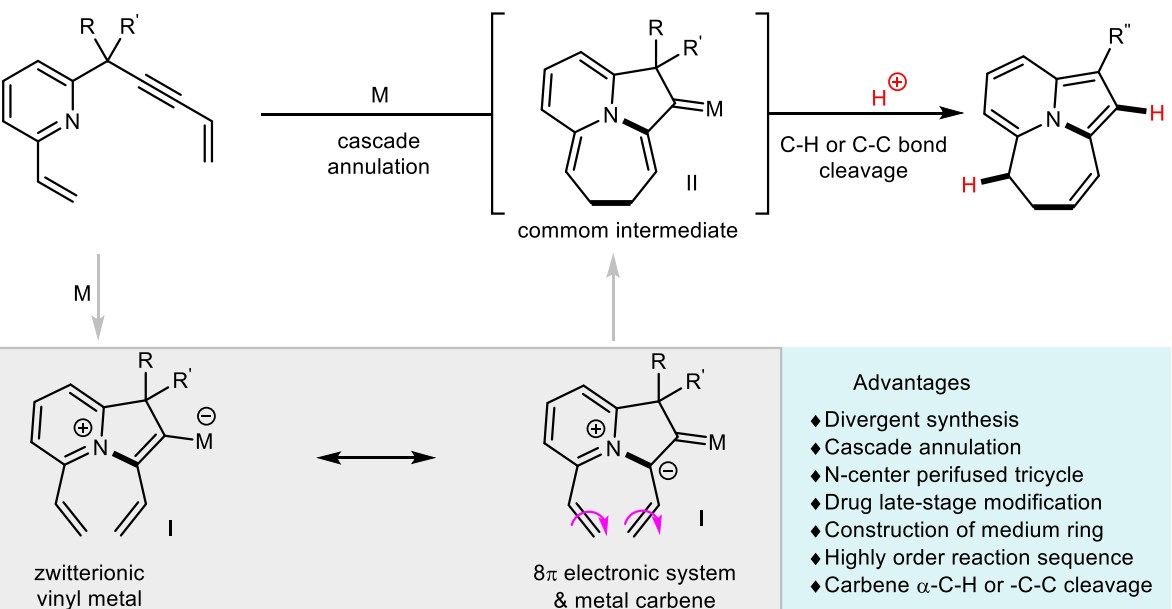

M
cascade
annulation

commom intermediate

C-H or C-C bond
cleavage

M

zwitterionic
vinyl metal

$8\pi$ electronic system
& metal carbene

Advantages

♦ Divergent synthesis
♦ Cascade annulation
♦ N-center perifused tricycle
♦ Drug late-stage modification
♦ Construction of medium ring
♦ Highly order reaction sequence
♦ Carbene $\alpha$-C-H or -C-C cleavage

**Fig. 1 | Background and our proposal. a** Representative tricarbocycles natural products. **b** Carbon atom vs nitrogen atom. **c** Traditional methodology of synthesis the NCPC. **d** Our strategy to build NCPC.

## Results

### Optimization of reaction conditions

We used the propargylic derivative of 6-vinylpyridine **1a** as our model substrate for cleaving α-C–H triggered by metal carbene (Table 1). Initially, we screened various metals based on the known alkynophilicity of gold[50,51], silver[52,53], and copper compounds[54–56]. When the solution of **1a** (0.05 mmol) in toluene was reacted using chloro(triphenylphosphine) gold (AuPPh₃Cl) as the catalyst at 90 °C, the expected NCPC product **2a** was formed with a 31% yield (Table 1, entry 1), and its structure was confirmed by X-ray single crystal diffraction analysis. Because gold and silver salts combine to release highly active gold species, improving the catalytic ability of the gold catalyst and then combining AuPPh₃Cl and AgOTf showed high reactivity and significantly improved the yield to 60% (Table 1, entry 2). Switching Au(I) to various silver salts, such as AgSbF₆, AgOTf, and Ag₃PO₄ yielded only trace amounts of the product (Table 1, entries 3–5). In contrast, various copper salts yielded the expected products in high yields in all cases (Table 1, entries 6–9). Among them, CuTc was used to produce the cyclization product with the highest yield (96%) (Table 1, entry 9). Subsequently, decreasing the reaction temperature from 90 °C to 60 °C decreased the yield (Table 1, entry 9). Finally, applying commonly used solvents, such as 1,2-DCE, THF and DMF, did not have a better effect (Table 1, entries 10–12).

### Reaction scope of Cu-catalyzed α-C–H bond cleavage

After establishing the optimized reaction conditions, we investigated the scope of NCPCs under Cu-catalysis (Fig. 2). The eneyne moieties possessing different aryl (**2a–2f**), heteroaryl (**2g** and **2h**), alkenyl (**2i**), and alkyl (**2j** and **2k**) substituents at the end of the alkene underwent smooth conversion to yield the corresponding NCPCs with excellent yields. Adding a substituent at the internal position of the eneyne moieties (**2l–2n**) also produced the desired products in high yields. The generality of this transformation was further extended by placing various substituents, including aryl (**2o–2s**), heteroaryl (**2t**), alkenyl (**2u**), and alkyl (**2v** and **2w**) substituents, on the alkenyl segment at the C6 position of pyridine, where these substrates could be converted into the expected products in high yields. Halogen-containing aryl-substituents (**2e, 2m** and **2s**) were also competent in the cascade cyclization process, providing an additional functional handle for further modification.

We further investigated the generality of this transformation by using the annular eneynes that were efficiently converted into the corresponding tetracyclic products (**2x** and **2y**) with excellent yields. When the annular eneyne was replaced by thiophene, the transformation was successful irrespective of whether the α-position or β-position of thiophene was used. Through this process, tetracycle products were formed with a fused aromatic ring (**2z** and **2aa**). The O- or N-containing aromatic compounds, such as benzofuran and indole, also underwent the cascade cyclization sequence, thus providing access to fused pentacyclic scaffolds (**2ab** and **2ac**). To confirm the structure of the products, a single crystal X-ray structure of product **2z** was obtained, and the structure of other products was deduced from their NMR spectra. In addition, we carried out this cascade process by using benzene ring as π-component. To our disappointment, the reaction of the phenyl group as the π-component was interrupted during the 5-endo-cyclization step, and subsequent 8π electro-cyclization of benzene did not proceed as predicted (please see Supplementary Information, Page 28). It may be attributed to the weak aromatic system of the electron-rich furan or thiophene due to uneven charge distribution. The inherent strong aromatic property of benzene makes it difficult to dearomatize in 8π electrocyclization process, resulting that the cascade sequence was interrupted in the 5-endo-cyclization step.

A part of this study involved research on the α-C–C bond cleavage triggered by metal carbene, in which the highly versatile carbocation

intermediate was formed and subsequently reacted with various nucleophilic reagents, leading to the formation of various nucleophilic substituted NCPC products. The three-membered cycle showed high structural strain[57,58], which provided a dynamic momentum and a thermodynamic driving force for α-C–C cleavage. Using this, the substrate bearing a cyclopropyl between pyridine and alkyne (**3a**) was synthesized, and the corresponding reaction was conducted to determine whether the ring-opening nucleophilic substitution reaction would occur through the α-C–C cleavage at the α-position of metal carbene. To our delight, we found that the cascade sequence also formed the expecting product (**4a**) with an 11% yield using CuTc as the catalyst and NaOAc as the nucleophilic reagent. In principle, the success of this divergent synthesis process depended on subtle differences in the reactivity between the starting material and the metal catalyst because both metal carbene of different reactivity triggered different transformations. We focused mainly on screening the metal catalyst to further improve the yield. After extensive screening, we found that the silver salts highly stimulated the α-C–C cleavage. Finally, using silver phosphate as the catalyst and TEBA as an additive in the presence of NaOAc, product **4a** was formed with an 80% yield (Detailed conditions screening, please see Supplementary Information, Supplementary Table 1). In general, the metal carbene triggered the nucleophilic ring-opening process of the three-membered ring, which resulted in the complete release of ring strain. This process is thermodynamically favorable compared to the formation of a four-membered ring through 1,2-alkyl migration.

### Reaction scope of Ag-catalyzed α-C–C bond cleavage

Next, we investigated the scope of the cascade sequence (Fig. 3). When the phenyl at the end of the eneyne was replaced with thienyl, the ring-opening reaction proceeded slowly to generate the thienyl-substituted product (**4b**) with an 89% yield. Besides aromatic rings, the alkenyl (**4c**) or alkyl (**4d** and **4e**) substitutions at the same position were also suitable substrates, affording the corresponding products in high yields. Similarly, the C2 and C3 positions of thiophene could be changed to obtain the desired tetracyclic products (**4f** and **4g**) with yields of 69% and 78%, respectively. The conversion of thiophene into indole was tolerated and led to the formation of the pentacyclic product (**4h**) with a 71% yield. When the cyclopropyl with a 4-Me-phenyl (**4i**) or 4-F-phenyl (**4j**), was exposed to the standard reaction condition to determine the effect of the substituents, the results indicated that the latter led to a lower yield probably because the para-F-substitution was not favorable for the stability of the carbocation intermediate. Additionally, using sodium formate as a nucleophilic reagent also generated the desired product (**4k**) with a satisfactory yield. Although the nucleophilicity of phenoxyl was relatively weak, it led to the corresponding oxo-substitution of NCPC (**4l**) with a 52% yield.

Subsequently, the reaction was conducted to test whether the transformation could be performed using different types of nucleophilic reagents. The results indicated that 2-thiol-benzothiazole as a nucleophilic reagent could produce the corresponding product (**4m**) with a 63% yield. The nitrogen-containing nucleophiles, such as aniline, sodium azide, 1,2,3-triazole, and tetrazole, were also investigated, and the results indicated that all nucleophilic reagents reacted effectively under standard conditions, resulting in the formation of the desired products (**4n–4q**) with satisfactory yields. Regarding the nucleophilicity of the carbon atom, we used the sodium salt of malononitrile and indole as nucleophilic reagents. Both substrates participated in this intermolecular ring-opening substitution reaction, forming the desired products (**4r** and **4s**) with satisfactory yields.

The late-stage modification of pharmaceuticals[59,60] is of significant importance for the discovering drugs or improving bioactivity (Fig. 4). To further demonstrate the potential of this strategy, we examined the application of the ring-opening nucleophilic substitution in the

**Table 1 | Optimization of the reaction conditions[a,b]**

| Entry | Catalyst | Solvents | T (°C) | Yield (%) |
|---|---|---|---|---|
| 1 | AuPPh₃Cl | PhMe | 90 | 31 |
| 2 | AuPPh₃Cl + AgSbF₆ | PhMe | 90 | 60 |
| 3 | AgSbF₆ | PhMe | 90 | Trace |
| 4 | AgOTf | PhMe | 90 | Trace |
| 5 | Ag₃PO₄ | PhMe | 90 | Trace |
| 6 | CuSO₄ | PhMe | 90 | 75 |
| 7 | Cu(OAc)₂ | PhMe | 90 | 84 |
| 8 | Cu(CH₃CN)₄PF₆ | PhMe | 90 | 68 |
| 9 | CuTc | PhMe | 90 (60) | 96 (23)[c] |
| 10 | CuTc | 1,2-DCE | 90 | 65 |
| 11 | CuTc | THF | 90 | 45 |
| 12 | CuTc | DMF | 90 | 31 |

[a]All reactions of **1a** (0.05 mmol, 1.0 equiv.) and catalyst (0.005 mmol, 10 mol%) were performed in solvent (1.0 mL) at 90 °C for 0.5 h.
[b]Isolated yield.
[c]Reaction was performed at 60 °C.

**Fig. 2 | Substrate scope of α-C–H bond cleavage.** Reaction Conditions: **1** (0.05 mmol) and CuTc (0.005 mmol) in toluene (1.0 mL) at 90 °C for 0.5 h. [a]R$^1$ = OTBS, [b]R$^1$ = OAc. Isolated yields are provided.

late-stage modification of biologically active small molecules. The different marketed pharmaceuticals, including structurally complicated Gemfibrozil, Idometacin, Estrone, Riluzole, and Candesartan cilexetil, accessing divergent products (**4t**−**4x**) with satisfactory yields.

These modifications can also provide key advantages from the perspective of medicinal chemistry. The catalytic system showed high compatibility with several nucleophilic reagents and rapidly established molecular complexity and diversity.

**Fig. 3 | Substrate scope of α-C−C bond cleavage.** Reaction Conditions: **2** (0.1 mmol), Ag₃PO₄ (0.02 mmol), TEBA (0.2 mmol) and nucleophile (0.3 mmol) in CHCl₃ (2.0 mL) at 100 °C for 24 h. Isolated yields are provided. TEBA: Benzyltriethylammonium chloride.

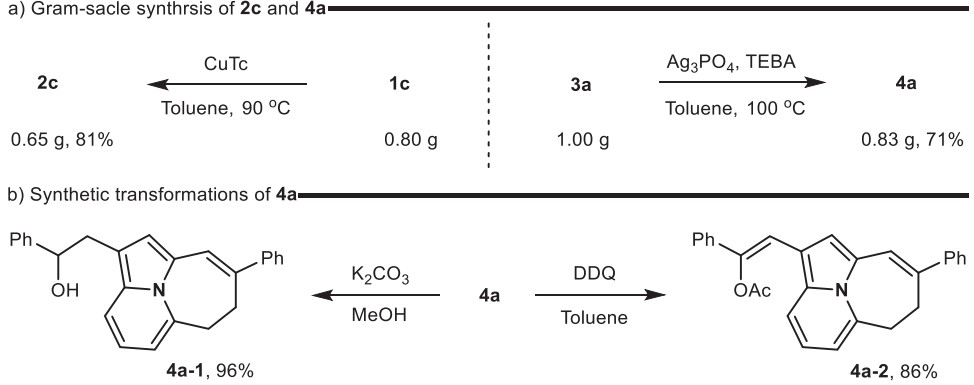

**Fig. 4 | Late-stage modification of pharmaceuticals.** Introducing the 5,6,7-perifused tricycle into pharmaceuticals through oxo or aza-nucleophilic ring-opening of α-metal carbene cyclopropane.

**Fig. 5 | Synthetic utility. a** Gram-scale synthesis. **b** Synthetic transformations.

## Synthetic utility and mechanistic studies

To demonstrate the practicability of our methodology, the gram-scale reactions of **1c** and **3a** were conducted, and the products **2c** and **4a** were formed with yield of 81% and 71% (Fig. 5a), respectively. Additionally, the hydrolysis of product **4a** occurred in the presence of potassium carbonate to deliver **4a-1** with a 96% yield. Next, oxidative dehydrogenation proceeded sluggishly; DDQ (1,2-dihloro-4,5-dicyanobenzoquinone) was used as an oxidant, which allowed the conversion of **4a** into **4a-2** rather than cycloazine (Fig. 5b). This selective oxidation can be attributed to the difference in C–H bond energy. The

α-position of the oxygen atom typically has a lower C–H bond energy compared to a normal C–H bond[61,62], making it more susceptible to radical generation during the oxidation by DDQ. Additionally, the tertiary radical on the side chain is more stable than the secondary radical on the tricyclic skeleton.

To understand this mechanism and elucidate some specific processes of the reaction, several control experiments were conducted. The deuterated starting material **1a'** was exposed to the standard conditions. The NMR spectra showed that the reaction generated product **2a** rather than the deuterated product **2a'** (Fig. 6a), indicating

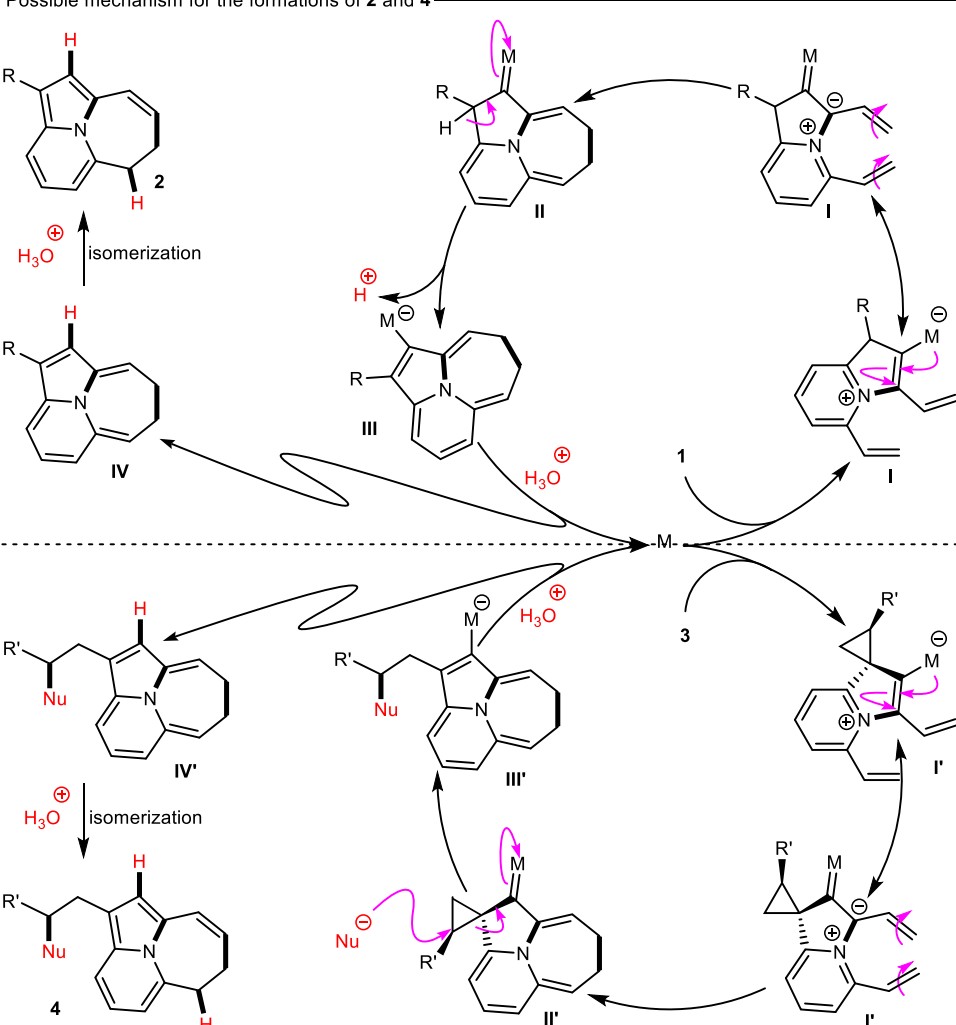

**Fig. 6 | Control experiments and possible mechanism. a** Control experiment of substrate **1a'**. **b** Control experiment of $D_2O$ as mixed solvent. **c** Possible mechanism for the formation of NCPC.

that the intramolecular 1, 2-hydride shift triggered by metal carbene was not involved in the process. To identify the H source, the non-deuterated substrate **1a** was reacted with a mixture of toluene with $D_2O$ (toluene/$D_2O$ = 1/0.2), and the reaction produced the deuterated

product **2a"**; the C4 and C2' positions were deuterated at a 90% and 100% deuterated ratio, respectively (Fig. 6b). These results suggested that the reaction underwent protonation twice in the presence of the proton donor that came from $H_2O$ in the solvent.

Based on the above mechanistic investigations and the results of our experiments, we proposed a plausible catalytic cycle (Fig. 6c). The 5-endo-cyclizations of substrates **1** and **3** proceeded under metal catalysis to form the intermediates **I** and **I′**, which formed the common metal carbene intermediates (**II**) and (**II′**), respectively, via 8π electrocyclization. The results of our control experiment showed that intermediate **II** triggered deprotonation through the cleavage of C–H to produce vinyl metal intermediate **III**, which allowed protonation in the presence of $H_2O$ to produce intermediate **IV** along with the release of the metal catalyst. Intermediate **II′** underwent ring-opening via a C–C bond cleavage, which involved nucleophilic substitution along with the generation of vinyl metal intermediate **III′**. A similar protonation process formed intermediate **IV′**. These two polyene intermediates **IV** and **IV′** underwent further isomerization to form final products **2** and **4**, respectively. The whole reaction underwent intermolecular protonation twice rather than intramolecular H-migration.

In conclusion, we showed the unique synthetic potential of cascade cyclization for constructing structurally diverse N-center polycycle skeletons. The key feature of this transformation was that the 8π electrocyclization could be used to construct the medium ring along with the generation of NCPC metal carbene, which served as versatile building blocks for α-C–H or -C–C cleavage. This procedure is an effective way to rapidly access structurally divergent nitrogen-containing polycyclic products. The 8π electrocyclization might also be used to develop efficient strategies for rapidly constructing a library of fused polycycles. These structurally attractive NCPC skeletons might be interesting since they can help further enhance the utility of this powerful synthetic strategy.

## Methods

### General procedure for α-C–H bond cleavage

The freshly prepared **1** (0.05 mmol, 1.0 equiv), CuTc (10.0 mol%), and toluene (1.0 mL) were added to a 5.0 mL round bottom flask at room temperature, the reaction mixture was stirred at 90 °C for 0.5 h and then cooled to room temperature. After completion, the reaction mixture was filtered, and the organic solution was concentrated under vacuo, residue was purified by column chromatography on alkaline alumina to afford the desired product.

### General procedure for α-C–C bond cleavage

$Ag_3PO_4$ (20.0 mol%), TEBA (0.2 mmol, 2.0 equiv) and nucleophilic reagent (0.3 mmol, 3.0 equiv) were added to a solution of freshly prepared **3** (0.1 mmol, 1.0 equiv) in $CHCl_3$ (2.0 mL) for a Schlenk tube at room temperature, the reaction mixture was stirred at 100 °C for 24 h and then cooled to room temperature. After completion, the reaction mixture was filtered, and the organic solution was concentrated under vacuo, residue was purified by column chromatography on silica gel to afford the desired product.

## Data availability

The authors declare that all data generated in this study are available within the paper and its Supplementary Information files. The X-ray crystallographic coordinates for structures reported in this article have been deposited at the Cambridge Crystallographic Data Center (**2a**: CCDC 2141700; **2z**: CCDC 2141701; **3a**: CCDC 2239536; **4 m**: CCDC 2141703). These data can be obtained free of charge from The Cambridge Crystallographic Data Centre via www.ccdc.cam.ac.uk/data_request/cif. Any additional detail can be requested from the corresponding authors.

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

## Acknowledgements

The authors thank NSFC (Grant 21772019 to B.L.), acknowledge the support from the Venture & Innovation Support Program for Chongqing Overseas Returnees (cx2019007 and cx2020047 to B.L.), and the Basic and Frontier Research Project of Chongqing (CSTB2022NSCQ-MSX0320 to B.L.) and the Analytical and Testing Centers of Chongqing University are gratefully acknowledged for instrumental facilities.

## Author contributions

J.H. performed most of the experiments, including the synthesis of substrates, collection of the data, mechanistic studies and analyzed the data. Y.Y. and Y.G. helped synthesize some of the substrates. X.T. and Y.T. repeated the results. B.L. conceived the study and wrote the paper. All authors discussed the results and approved the final version of the manuscript.

## Competing interests

The authors declare no competing interests.
