## [Peer Review File · Nature Communications]

Divergent Access to 5,6,7-perifused CyclesReviewers' Comments:

Reviewer #1:

Remarks to the Author:

The preparation of N-center perifused cycles is a long-term challenge in organic synthesis. Li and coworkers have reported a transition metal-catalyzed high-order reaction sequence for synthesizing a structurally unique N-center 5,6,7-perifused cycle. This is a highly impressive work. This manuscript provides a practical and efficient strategy for the preparation of structurally unique and novel nitrogen-containing heterocycle, including some interesting tetra- and pentacyclic N-center 5,6,7-perifused cycle skeletons. Based on my own knowledge in this field, this is a breakthrough progress. The reactions generally proceed with high-order cascade sequence, avoiding the cumbersome multi-step process and presenting a diverse range of substrates and good functional group tolerance. Benzene as π -component are not good functional groups in typical electrocyclization reactions. In most of the reported electrocyclization reactions, the electron-rich aromatics are far more reactive than those benzene as well as their analogues. However, I suggest the authors do additional trials to investigate the possibilities for the involvement of benzene. Typically, the metal carbene intermediate may also trigger intramolecular 1,2-alkyl migration. In this reaction the ring enlargement products by the releasing of the strain of three membered ring were detected? These results will be important and should be discussed in the manuscript.

In general, this is a high-quality manuscript that provides a high-order reaction sequence for efficient and divergent access to N-center 5,6,7-perifused cycle skeletons. The gram-scale level and the late-stage modifications of pharmaceuticals are well performed. And the mechanism is relatively clear from some control experiments. The supporting information is also thorough and provides all the expected data. Considering the novelty and the importance of this work, I recommend it publishing in nature communications after minor revisions.

Some additional questions:

1. Some mistakes should be corrected in text and SI. For example, the "wittig reagent" should be corrected into "wittig reagent" in SI.
2. Could the metal carbene be directly captured by nucleophilic reagents when the propargyl position is a quaternary carbon rather than a three-membered ring?
3. For the deuterated experiment of 1a', Where has deuterium atom gone? The author should give some explanation.

Reviewer #2:

Remarks to the Author:

The manuscript described by Li and coworkers reported an interesting cascade annulation reaction to access a series of N-center perifused multicycles compounds. The key features are the formation of medium by an unusual 8π electrocyclization reaction to achieve final fused tricycle system, overcoming the issue of medium-size and multi-fused cycle ring-strain. The final tricycle skeletons could be constructed by the both α -C-H or β -C-C cleavage initiated by metal a complicated carbene intermediate. Therefore, it should be acceptable after minor revisions.

1. What is the diastereoisomer ratio for the substrate 3? Is it a single diastereoisomer? Its relative configuration and dr value should be confirmed.
2. When the NaOAc were used as nucleophilic reagents, the NaOH as a side product should be generated. It may be unamiable for reaction time or the yield. Have you tried acetic acid as nucleophilic reagent.
3. The DDQ oxidized side-chain rather than the single bond of core tricycle skeleton. What reason causes this chemoselectivity? The result should be discussed in the manuscript.
4. Water as proton donor participated in the reaction process. How to control the amount of water? The detail should be described in the manuscript or SI.
5. The heteroaromatic ring as a π -component participated in the final cyclization. Can benzene ring participate in the cyclization process? The corresponding reaction should be investigated and the result

should be putted in manuscript.

6. The title of figure 4a need be revised, because it is not a deuterium experiment but a reaction of deuterated substrate.

Point-by-point responses to reviewers' comments

Response to comments of Reviewer # 1

Recommendation: publish after minor revisions.

Response: We greatly appreciate your insightful comments and suggestions.

Comment 1: Benzene as π -component are not good functional groups in typical electrocyclization reactions. In most of the reported electrocyclization reactions, the electron-rich aromatics are far more reactive than those benzenes as well as their analogues. However, I suggest the authors do additional trials to investigate the possibilities for the involvement of benzene.

Response: Based on your suggestions, we performed the transformation by using benzene as the π -component, as evidenced by the NMR spectra provided below. Supplementary experiments revealed that the reaction of the phenyl group as the π -component was interrupted during the 5-endo-cyclization step, and subsequent 8π electrocyclization of benzene did not proceed as predicted. We have added this information in our revised manuscript on Page 2: "It may be attributed to the weak aromatic system of the electron-rich furan or thiophene due to uneven charge distribution. The inherent strong aromatic property of benzene makes it difficult to dearomatize in 8π electrocyclization process, resulting that the cascade sequence was interrupted in the 5-endo-cyclization step." Additionally, the corresponding data has been added to our revised Supplementary Information (on pages S17, S28, S73, S118).

Supplementary experiment:

Comment 2: Typically, the metal carbene intermediate may also trigger intramolecular 1,2-alkyl migration. In this reaction the ring enlargement products by the releasing of the strain of three membered ring were detected? These results will be important and should be discussed in the manuscript.

Response: We did not detect any 1,2-alkyl migration products in all of our reactions. Instead, the metal carbene triggered the nucleophilic ring-opening process of the three-membered ring, which resulted in the complete release of ring strain. This process is thermodynamically favorable compared to the formation of a four-membered ring through 1,2-alkyl migration. There are numerous examples illustrating this phenomenon (Selected examples, *J. Am. Chem. Soc.* **122**, 11549–11550 (2000); *J. Am. Chem. Soc.* **123**, 10511–10520 (2001); *Chem. Soc. Rev.* **33**, 431–436 (2004)). We have included a corresponding description in our revised manuscript on Page S3.

Comment 3: Some mistakes should be corrected in text and SI. For example, the "wittig reagent" should be corrected into "wittig reagent" in SI.

Response: We apologize for the oversight. We have corrected the term "wittig reagent" to "wittig reagent" in our revised Supplementary Information (pages S3, S5).

Comment 4: Could the metal carbene be directly captured by nucleophilic reagents when the propargyl position is a quaternary carbon rather than a three-membered ring?

Response: We thank this referee for the valuable suggestions. We prepared the pyridine substrates with a quaternary carbon at the α -position (as indicated by the NMR spectra provided below). Both the substrates bearing phenyl and methyl groups were tested in our reactions. However, we did not observe any products under the standard reaction conditions, with the starting materials recovered in good yields. We have added the corresponding data in our revised Supplementary Information (pages S7, S17, S74, S75).

Supplementary experiments

Comment 5: For the deuterated experiment of 1a', Where has deuterium atom gone? The author should give some explanation.

Response: Based on our control experiments, it was observed that the metal carbene induced α -C-D cleavage of the intermediate **II** can lead to the formation of a vinyl metal intermediate **III**, with a subsequent release of the deuterium atom within this step (as depicted in the figure below). A detailed description of this process has been presented in the mechanism discussion in our manuscript as depicted below.

Response to comments of Reviewer # 2

Recommendation: publish after minor revisions.

Response: We greatly appreciate your positive and relevant comments.

Comment 1: What is the diastereoisomer ratio for the substrate 3? Is it a single diastereoisomer? Its relative configuration and dr value should be confirmed.

Response: We did not find the formation of the diastereoisomer during the preparation of the substrates, and the relative configuration was confirmed by the X-ray structure analysis of substrate **3a**. Consequently, we have included the relevant description in our revised Supplementary Information on Page S6.

Comment 2: When the NaOAc were used as nucleophilic reagents, the NaOH as a side product should be generated. It may be unamiable for reaction time or the yield. Have you tried acetic acid as nucleophilic reagent?

Response: In addition to using NaOAc as the nucleophilic reagent to initiate α -C-C bond cleavage under our standard conditions, we also conducted the reaction using HOAc. However, we observed that HOAc resulted in a lower yield of the desired product compared to NaOAc. We have included this information in our revised Supplementary Information on Pages S7 and S28.

Supplementary experiment

Comment 3: The DDQ oxidized side-chain rather than the single bond of core tricycle skeleton. What reason causes this chemoselectivity? The result should be discussed in the manuscript.

Response: This selective oxidation can be attributed to the difference in C-H bond energy. The α -position of the oxygen atom typically has a lower C-H bond energy compared to a normal C-H bond (ref. *Russ. Chem. Rev.* **74**, 825–858 (2005); *Russ. Chem. Bull.* **51**, 1641–1650 (2002)), making it more susceptible to radical generation during the oxidation by DDQ. Additionally, the tertiary radical on the side chain is more stable than the secondary radical on the tricyclic skeleton. We have included corresponding discussions on this selective oxidation in our revised manuscript on Page 4.

Russ. Chem. Rev. 74, 825-858 (2005); *Russ. Chem. Bull.* 51, 1641-1650 (2002)

Comment 4: Water as proton donor participated in the reaction process. How to control the amount of water? The detail should be described in the manuscript or SI.

Response: Thank you for your suggestion. We determined the water content in toluene ($\text{H}_2\text{O} \approx 0.30\%$) and chloroform ($\text{H}_2\text{O} \approx 0.35\%$) using the Karl Fischer Coulomb titration method (ref. *Drug Dev. Ind. Pharm.* **51**, 1891–1903 (1988)). We have included a corresponding description of the water content in our revised Supplementary Information on Page S1.

Karl Fischer Moisture Titrator (Volumetric Method)

$$M\% = (W_1 \times V_2 \times 100\%) \div (W_2 \times V_1)$$

M: Mass fraction of water in organic solvent

W_1 : Quality of water in the blank experiment

W_2 : Quality of organic solvent

V_1 : Volume of Fischer reagent consumed in blank experiment

V_2 : Volume of Fischer reagent consumed by organic solvent

Toluene: $W_1 = 0.1800$ g, $W_2 = 2.7152$ g, $V_1 = 3.12$ ml, $V_2 = 0.14$ ml

$$M_{\text{Toluene}\%} = (0.1800 \text{ g} \times 0.14 \text{ mL} \times 100\%) \div (2.7152 \text{ g} \times 3.12 \text{ mL}) \approx 0.30\%$$

CHCl_3 : $W_1 = 0.1800$ g, $W_2 = 3.9011$ g, $V_1 = 3.12$ ml, $V_2 = 0.24$ mL

$$M_{\text{CHCl}_3\%} = (0.1800 \text{ g} \times 0.24 \text{ mL} \times 100\%) \div (3.9011 \text{ g} \times 3.12 \text{ mL}) \approx 0.35\%$$

Comment 5: The heteroaromatic ring as a π -component participated in the final cyclization. Can benzene ring participate in the cyclization process? The corresponding reaction should be investigated and the result should be putted in manuscript.

Response: This question is similar to Comment 1 raised by Reviewer 1, and we have provided a detailed response in our reply to Reviewer 1's Comment 1. Please refer to Page 2 of this letter for detailed information.

Comment 6: The title of figure 4a need be revised, because it is not a deuterium experiment but a reaction of deuterated substrate.

Response: We have modified the title of Figure 4a in our revised manuscript (page 4) and Supplementary Information (page S9) from "Deuterated experiment of substrate 1a" to "Control experiment of substrate 1a".

Reviewers' Comments:

Reviewer #1:

Remarks to the Author:

Authors have revised the manuscript according to the comments. I recommend the acceptance of this work.

Reviewer #2:

Remarks to the Author:

The authors have addressed all of my concerns. In my opinion, the manuscript can be accepted for publication.